# The Involvement of Oxidative Stress in Psoriasis: A Systematic Review

**DOI:** 10.3390/antiox11020282

**Published:** 2022-01-29

**Authors:** Elena-Codruța Dobrică, Matei-Alexandru Cozma, Mihnea-Alexandru Găman, Vlad-Mihai Voiculescu, Amelia Maria Găman

**Affiliations:** 1Doctoral School, University of Medicine and Pharmacy of Craiova, 200349 Craiova, Romania; 2Department of Dermatology, “Elias” University Emergency Hospital, 011461 Bucharest, Romania; 3Faculty of Medicine, “Carol Davila” University of Medicine and Pharmacy, 050474 Bucharest, Romania; matei-alexandru.cozma@drd.umfcd.ro (M.-A.C.); mihneagaman@yahoo.com (M.-A.G.); 4Department of Gastroenterology, Colentina Clinical Hospital, 020125 Bucharest, Romania; 5Department of Hematology, Center of Hematology and Bone Marrow Transplantation, Fundeni Clinical Institute, 022328 Bucharest, Romania; 6Department of Pathophysiology, University of Medicine and Pharmacy of Craiova, 200349 Craiova, Romania; gamanamelia@yahoo.com or; 7Clinic of Hematology, Filantropia City Hospital, 200143 Craiova, Romania

**Keywords:** psoriasis, oxidative stress, reactive oxygen species, antioxidants, inflammation

## Abstract

Psoriasis is a chronic, immune-mediated inflammatory dermatosis characterized by the appearance of erythematous plaques, covered by white scales, occasionally pruritogenic, and distributed mainly on the extensor areas. Oxidative stress is defined as an imbalance or a transient or chronic increase in the levels of free oxygen/nitrogen radicals, either as a result of the exaggerated elevation in their production or the decrease in their ability to be eliminated by antioxidant systems. Although the pathogenesis of psoriasis remains far from elucidated, there are studies that delineate an involvement of oxidative stress in this skin disorder. Thus, a systematic search was computed in PubMed/Medline, Web of Science and SCOPUS and, in total, 1293 potentially eligible articles exploring this research question were detected. Following the removal of duplicates and the exclusion of irrelevant manuscripts based on the screening of their titles and abstracts (n = 995), 298 original articles were selected for full-text review. Finally, after we applied the exclusion and inclusion criteria, 79 original articles were included in this systematic review. Overall, the data analyzed in this systematic review point out that oxidative stress markers are elevated in psoriasis and share an association with the duration and severity of the disease. The concentrations of these biomarkers are impacted on by anti-psoriasis therapy. In addition, the crosstalk between psoriasis and oxidative stress is influenced by several polymorphisms that arise in genes encoding markers or enzymes related to the redox balance. Although the involvement of oxidative stress in psoriasis remains undisputable, future research is needed to explore the utility of assessing circulating serum, plasma, urinary and/or skin biomarkers of oxidative stress and of studying polymorphisms in genes regulating the redox balance, as well as how can these findings be translated into the management of psoriasis, as well in understanding its pathogenesis and evolution.

## 1. Introduction

Psoriasis is a chronic, immune-mediated inflammatory dermatosis characterized by the appearance of erythematous plaques, covered by white scales, occasionally pruritogenic, and distributed mainly on the extensor areas (elbows, knees, scalp, chest) [1]. It is one of the most common chronic dermatological diseases, with a worldwide prevalence and incidence that varies significantly depending on geographical area, age and sex. For example, its prevalence varies from 0–2.1% in the pediatric population to 0.91–8.5% in adults, while its incidence is 40.8 cases in 100,000 people in the former and 78.9–230 cases in 100,000 in the latter [2]. Psoriasis is a debilitating condition, which significantly affects the quality of life and impacts on the life expectancy of individuals who suffer from it. Subjects diagnosed with this dermatological disorder experience feelings of depression, social stigma, and frequently associate other chronic systemic diseases, e.g., cardiovascular disease, obesity, diabetes, psoriasis no longer being considered a condition which strictly involves the integument [3,4]. The pathophysiology of the disease is still incompletely elucidated and multiple factors are said to be involved in its initiation and perpetuation, e.g., stressors, genetic factors, environmental factors (air pollutants, cigarette smoke), etc. [5]. In the last decade, a new theory is trying to explain the pathophysiology of this condition by looking at the role played by oxidative stress and chronic inflammation in the initiation of keratinocyte proliferation and differentiation that underline psoriasis [6]. Oxidative stress is defined as an imbalance or a transient or chronic increase in the levels of free oxygen/nitrogen radicals, either as a result of the exaggerated elevation in their production or the decrease in their ability to be eliminated by antioxidant systems [7]. Due to its role as a barrier and its direct exposure to environmental factors, the skin is an important source of free radicals that play, when in low concentrations, an essential role in the defense against microorganisms and in cell differentiation [8]. When their concentration increases, leading to oxidative stress, free radicals appear to be involved in DNA alteration, cell protein degradation, lipid oxidation, apoptosis, tissue injury, altered response of T-helper cells and secretion of interleukin-17 (IL-17) [9]. As all these are essential stages in the initiation and perpetuation of psoriasis, the hypothesis that oxidative stress plays a key role in the pathophysiology of this chronic dermatosis has emerged [10].

Thus, in order to evaluate the involvement of oxidative stress in psoriasis, we conducted the current systematic review. In order to investigate the scientific progress achieved so far in understanding the role of oxidative stress in this dermatological illness, we investigated the associations between different parameters of the antioxidant status, the total oxidative capacity and the severity of psoriasis, the role in psoriasis of different genetic polymorphisms in enzymes involved in the redox balance, as well as the influence of certain topical or systemic treatments routinely applied in psoriasis on the redox balance.

## 2. Materials and Methods

In the preparation of this manuscript, we followed the recommendations stated in the Preferred Reporting Items for Systematic Reviews and Meta-Analyses (PRISMA) guidelines [11]. Thus, three investigators (E.-C.D., M.-A.C., M.-A.G.) performed an advanced search in PubMed/Medline, Web of Science and SCOPUS using specific keywords and/or word combinations from the inception of these databases up to 23 July 2021. The inclusion criteria considered were: 1. Original studies evaluating the relationship between different parameters of oxidative status and psoriasis severity OR original studies evaluating different polymorphisms in genes encoding enzymes involved in the oxidative status in psoriasis OR original studies which assessed the oxidative status of patients with psoriasis before and after the use of approved topical or systemic antipsoretic therapies. 2. The selected original studies were conducted in the adult population (study subjects aged ≥ 18 years). 3. The selected articles were written in a language spoken by the authors (English, French, Italian or Romanian). 4. The full-text of the selected articles was available for retrieval. 5. The studies were published on/after 1 January 2000. The exclusion criteria considered were represented by: 1. Studies performed in children, on cell cultures or on animals. 2. Studies whose full-text version was written in a language other than the aforementioned ones. 3. Reviews, letters to the editor, case reports or abstracts presented at scientific events. 4. The full-text version of the article was unavailable. 5. Studies did not report sufficient information on the role of oxidative stress in psoriasis, on polymorphisms of genes encoding enzymes involved in regulating oxidative stress or on oxidative stress levels before and after the use of local or systemic psoriasis therapies. 6. The studies were published before 1 January 2000. Any disagreement between the investigators was resolved by consultation with the senior coordinators (M.-V.V. and A.-M.G.) of the project, allowing for the final selection of the papers to be included in this systematic review. We evaluated the methodological quality and the risk of bias of the analyzed studies using the methodological index for non-randomized observational studies (MINORS) and the Mixed Methods Appraisal Tool (MMAT), respectively [12,13]. This protocol was registered in PROSPERO (ID 306997).

## 3. Results

The flow diagram of the literature search process is reported in Figure 1. We detected a total of 1293 potentially eligible articles based on the aforementioned search strategy. Of these, after we removed the duplicates and the papers whose title and abstract did not correspond to the aim of this review (n = 995), 298 original articles were selected for full-text review. After we applied the inclusion and exclusion criteria, a total of 79 articles were included in the present systematic review.

### 3.1. Markers of Oxidative Stress in Patients with Psoriasis

A total of 53 studies evaluated the following markers or parameters of oxidative stress in patients with psoriasis and also the correlations between those markers and severity and duration of disease: catalase (CAT), myeloperoxidase (MPO), ferroxidase (FOX), ischemia modified albumin (IMA), paraoxonase-1 (PON-1), total oxidant status (TOS), total antioxidant status (TAS), malondialdehyde (MDA), 8-hydroxy 2′-deoxyguanosine (8H2D), advanced oxidation protein products (AOPP), nicotinamide adenine dinucleotide phosphate oxidase (NADPH oxidase), reactive oxygen species (ROS), glutathione peroxidase (GSH-Px), glutathione (GSH), thioredoxin reductase (TrxR), ferric reducing ability of plasma (FRAP), arylesterase (AS), oxidative stress index (OSI), salivary peroxidase (Px), superoxide dismutase (SOD), advanced glycation end-products (AGE), lipid hydroperoxides (LOOH), oxidized LDL (OxLDL), nitric oxide (NO), 8-hydroxy guanosine (8-OHdG), inducible nitric oxide synthase (iNOS), glutathione reductase (GSH-R), 25-hydroxy vitamin D (25-OH-vitD), adenosine deaminase (ADA), thiobarbituric acid (TBA), total peroxide concentration (TPX), protein carbonyl compounds (PCC), pyrrolized protein (PP), oxidized glutathione (GSSG), total bilirubin (TB), direct bilirubin (DB), indirect bilirubin (IB), copper (Cu), iron (Fe), transferrin (Trf), autoantibodies anti-oxidized LDL (AuAb-oxLDL), interleukin-6 (IL-6), C-reactive protein (CRP), antibodies against carboxyethyllysine (Ab anti-CEL) and antibodies against carbocymethyllysine (Ab anti CML) [14,15,16,17,18,19,20,21,22,23,24,25,26,27,28,29,30,31,32,33,34,35,36,37,38,39,40,41,42,43,44,45,46,47,48,49,50,51,52,53,54,55,56,57,58,59,60,61,62,63,64,65,66]. Table 1 summarizes the information presented in these studies.

Kirmit et al. evaluated 87 patients with psoriasis in comparison with 60 healthy subjects and showed a significant increase in oxidative stress markers, namely catalase (*p* = 0.04), ferroxidase, myeloperoxidase and ischemia-modified albumin (*p* < 0.001 for all), in psoriasis versus the control group [14].

Ischemia-modified albumin appears to be a new marker of oxidative stress in this dermatological disease, as Ozdemir et al. also observed elevated levels of this molecule (*p* = 0.001) in patients with psoriasis versus controls. These findings confirm the theory that oxidative stress plays a key role in the development of this dermatosis and its complications, e.g., cardiovascular disease [48].

Kizilyel et al. (2019) assessed total oxidant status, total antioxidant status, malondialdehyde and 8-hydroxy 2′-deoxyguanosine levels in 95 Turkish patients (50 with psoriasis versus 45 controls) and detected a significant elevation of total oxidant status only (*p* < 0.001) [19]. Skoie et al. recorded no statistically significant changes in advanced oxidation protein products and malondialdehyde levels in psoriasis. However, the research showed a significant increase in these parameters in men compared to women (*p* < 0.01) [18]. In Yldirim et al. ’s study, malondialdehyde was not significantly higher in the serum of psoriatic patients versus controls; however, an increase in this enzyme was measured in the samples obtained by lesional skin biopsy [65].

Although patients with psoriasis have higher levels of total reactive oxygen species and superoxide ion levels in CD4+ T lymphocytes (*p* = 0.04), no significant differences were recorded in terms of psoriasis severity. In addition, there were no differences in the concentrations of intracellular glutathione and total antioxidant status between the two groups. However, in these patients, the researchers reported an association between the intracellular levels of reactive oxygen species and superoxide ions and glutathione levels in T lymphocytes (*p* = 0.03) [22].

Emre et al. analyzed concentrations of total oxidant status, total antioxidant status, arylesterase and oxidative stress index in smoker and non-smokers psoriasis patients versus controls with the same characteristics. The results consisted of a decrease in total oxidant status and an increase in total oxidant status and oxidative stress index in psoriasis patients, compared to the controls, without any differences between smokers and non-smokers in psoriasis [46]. Moreover, the levels of nitric oxide, as well as plasma and tissue malondialdehyde, were higher in patients with psoriasis (*p* = 0.001). The integument areas affected by psoriasis displayed elevated concentrations of tissue malondialdehyde versus the intact skin (*p* = 0.003) [52].

Ambrozewicz et al. assessed plasma levels of NADPH oxidase, xanthine oxidase, glutathione peroxidase, glutathione reductase, superoxide dismutase, thioredoxin reductase, glutathione and vitamin C in patients with psoriatic arthritis or psoriasis compared to the healthy population. In addition, the investigators concluded that, apart from an elevation in oxidative stress levels in the two aforementioned ailments, the expression of nuclear erythroid factor 2-related factor 2 (Nrf2) and the cannabinoid receptor type 2 was raised in patients with psoriasis vulgaris only, indicating that antioxidant and anti-inflammatory systems are overactive in these subjects [25]. Other elements involved in the oxidative balance are native and total thiols which have been ascertained by Emre et al. to be elevated in psoriasis [28].

Zhou et al. assessed, in a comparative study (214 psoriasis patients versus 165 healthy counterparts), total bilirubin and CRP values and discovered that, in psoriasis, there is a significant reduction in total bilirubin (*p* < 0.001) and an elevation in CRP (*p* < 0.001). Total bilirubin is associated with oxidative stress and chronic inflammation and predisposes individuals with psoriasis to atherosclerosis, regardless of the PASI (Psoriasis Area Severity Index) [35]. Balta et al. also observed lower levels of total bilirubin (*p* < 0.05), direct bilirubin (*p* < 0.001) and higher levels of indirect bilirubin (*p* < 0.05) and CRP (*p* < 0.001) in this dermatosis, emphasizing once again the association between total bilirubin, atherosclerosis and cardiovascular diseases (as systemic complications of psoriasis), as these subjects display an elevated carotid intima-media thickness [41]. The fact that both serum malondialdehyde and IL-6 levels are elevated in individuals with psoriasis and coronary heart disease emphasizes the involvement of oxidative stress in the pathogenesis of these afflictions [36]. Other markers of oxidative stress, relevant in the assessment of cardiovascular risk, are oxidized LDL, anti-oxidized LDL antibodies and the autoantibodies anti-oxidized LDL/oxidized LDL ratio, all of which are elevated in patients with psoriasis. Asha et al. also confirmed that oxidized LDL (*p* < 0.01) and malondialdehyde (*p* < 0.001) are elevated in psoriasis [27]. This solidifies the theory that increased levels of oxidative stress in this skin disease favor the onset of cardiovascular complications [32]. 

Notwithstanding, markers of oxidative stress have been detected in elevated concentrations not only in the serum of individuals with psoriasis, but also in their saliva or urine. Thus, research performed on 40 patients with this debilitating disorder versus an equal number of controls disclosed that total oxidant status, oxidative stress index, advanced glycation end-products, advanced oxidation protein products, malondialdehyde and lipid hydroperoxides are higher in both the blood and the (un)stimulated saliva of individuals with psoriasis (*p* < 0.001) [16]. Moreover, the levels of urinary biopyrins proved to be significantly increased in patients with psoriasis versus controls (*p* < 0.001), implying that these molecules may emerge as new biomarkers of oxidative stress in this skin disease [31]. Both nitrate and urinary 8-hydroxy 2′-deoxyguanosine levels can be used as markers of oxidative stress in patients with psoriasis or atopic dermatitis, as Nakai et al. detected higher values in patients with psoriasis compared to the control group [57]. Other potential markers for monitoring oxidative stress in mild-to-moderate psoriasis cases are advanced oxidation protein products, visfatin and nesfatin. The latter two molecules, secreted by the adipose tissue, the hypothalamus and the peripheral tissues, trigger oxidative stress via their pro-inflammatory actions and pro-oxidant effects on lipids. Elevated levels of visfatin and advanced oxidation protein products were observed in individuals with mild-to-moderate forms of psoriasis versus healthy counterparts, with no significant differences in mild versus moderate forms of the ailment [24]. Papagrigoraki et al. compared, in an investigation that recruited 80 psoriasis versus 40 eczema versus 40 control cases, respectively, advanced glycation end-products concentrations measured non-invasively on the skin and in the serum. Higher skin advanced glycation end-products levels were observed in psoriasis versus eczema (*p* = 0.02) and healthy cases (*p* = 0.01). The presence or absence of lesions on the integument of individuals with psoriasis or eczema did not influence advanced glycation end-products measurements [29]. In addition, advanced glycation end-products and anti-glycated residues antibodies (anti-CML/CEL antibodies) are markedly elevated in the active versus the remission phase of the disease and versus healthy controls (*p* < 0.05). Moreover, no significant differences were observed between these parameters in subjects in the remission phase and the healthy group, emphasizing the role of carbonyl stress in the pathogenesis of the active forms of the disease [45].

Abeyakirthi et al. analyzed of arginine and ornitine (the compound formed via the action of arginase on arginine) concentrations in eight patients with psoriasis in comparison with the same number of healthy counterparts. An elevated arginase activity results in decrease in nitric oxide levels in these patients in a paradoxical manner because competition for the same substrate, i.e., arginine, occurs between the two enzymes, namely arginase and nitric oxide synthase. Thus, elevated ornitine (*p* < 0.001) and decreased arginine (*p* = 0.005) values were noted in individuals with psoriasis, confirming the aforementioned theory [55].

In psoriasis, trace elements, such as Cu and Fe, as well as acute phase proteins, seem to display alterations dependent on oxidative stress concentrations. Thus, Shahidi-Dadras et al. observed a decrease in transferrin and Fe (*p* < 0.01), as well as an increase in ceruloplasmin (*p* = 0.02), in this skin disorder [30]. 

Patients with psoriasis have lower paraoxonase 1 (*p* < 0.001) and alpha tocopherol (*p* < 0.05) and increased uric acid (*p* < 0.05) and homocysteine (*p* < 0.001) levels versus their healthy counterparts, however these markers are not associated with disease severity [15]. Moreover, Emre et al. also noted that lower levels of paraoxonase 1 and arylesterase in psoriasis associated with tobacco consumption versus healthy non-smokers (*p* = 0.01) [46].

#### Associations of Oxidative Stress Markers with the Duration and Severity of Psoriasis

A study on 20 patients with mild-to-moderate psoriasis (according to the PASI score) showed a statistically significant negative correlation between thioredoxin reductase levels measured in biopsy samples obtained from skin lesions and psoriasis severity (r = −0.85, *p* < 0.01). There also appears to be a correlation between paraoxonase 1 levels and disease duration (*p* < 0.05) and between homocysteine levels and a negative family history for psoriasis (*p* < 0.05) [15]. Paraoxonase 1 was correlated with disease severity in the investigation of He et al. (r = −0.49, *p* = 0.01), as well as with indices of chronic inflammation, e.g., IL-6 (r = 0.46, *p* = 0.02) and hs-CRP (r = 0.39, *p* = 0.05) [43]. In addition, a statistically significant negative association was noted in patients with psoriasis without metabolic syndrome (r = −0.42, *p* = 0.02), a correlation that was not present in psoriasis patients with cardiometabolic alteration [50]. Similar results were presented by Ferretti et al. who observed lower concentrations of paraoxonase 1/arylesterase (*p* < 0.001) and higher lipid hydroperoxides (*p* < 0.001) in individuals with severe to moderate versus mild forms of psoriasis [51]. 

In terms of salivary biomarkers, the concentrations of total oxidant status and lipid hydroperoxides in unstimulated saliva and in plasma, respectively, correlate with the severity of the disease (PASI) (r = 0.5 and r = 0.54, respectively, and r = 0.63 and r = 0.66, respectively, *p* < 0.001 for all). Regarding the associations with disease duration, total antioxidant status (r = −0.51), oxidative stress index (r = 0.58), malondialdehyde (r = −0.58) from unstimulated saliva and total oxidant status (r = 0.52) were significantly correlated (*p* < 0.001) with the number of years of illness [16]. A new marker of oxidative stress, namely urinary biopyrins, was also correlated with age (r = 0.29, *p* = 0.01) and the severity of the disease (r = 0.99, *p* < 0.001) in patients with psoriasis [31]. Both the level of urinary nitrates (r = 0.71, *p* = 0.003) and urinary malondialdehyde (r = 0.49, *p* = 0.04) correlated with the severity of the disease measured using the PASI score [31]. Although advanced oxidation protein products and visfatin levels do not correlate with disease severity, they appear to be associated with each other (r = 0.6, *p* < 0.001), but also with ultrasound markers of atherosclerosis, i.e., such as carotid intima-media thickness (r = 0.3, *p* < 0.05 for advanced oxidation protein products) and flow-mediated dilatation (r = −0.25, *p* < 0.05 for advanced oxidation protein products; r = −0.25, *p* < 0.05 for visfatin) [24]. Another marker of elevated oxidative stress in this skin disorder and which is also involved in the development of comorbidities in these patients is the advanced glycation end-products. Ergun et al. showed that the level of advanced glycation end-products detected non-invasively on the skin using autofluorescence methods is positively correlated with the carotid intima-media thickness (r = 0.30, *p* = 0.04), but also with the BMI (r = 0.42, *p* = 0.007) [49]. Moreover, the non-invasive skin measurement of advanced glycation end-products correlates with its plasma levels in individuals with psoriasis (r = 0.93, *p* < 0.001), whereas its serum levels correlate with the severity of psoriasis (r = 0.91, *p* < 0.001) [29]. Total bilirubin levels also appear to be correlated with vascular ultrasound changes in psoriasis and the development of atherosclerotic disease. Consequently, Balta et al. observed a pronounced correlation of total bilirubin not only with PASI (r = −0.35, *p* = 0.005), but also with the carotid intima-media thickness (r = −0.39, *p* = 0.005) [41]. Both malondialdehyde and IL-6 concentrations are associated with intima-media thickness and decreased coronary flow values [36].

Moreover, it was observed that oxidized LDL levels differ based on the severity of the disease quantified by the PASI score. Oxidized LDL were detected in higher concentrations in severe versus mild–moderate cases (*p* < 0.001). Regarding antioxidant markers, the ferric reducing ability of plasma level was lower in patients with severe versus mild–moderate forms, but the gluthatione levels were similar [27]. Moreover, the ratio between autoantibodies anti-oxidized LDL and oxidized LDL was notably correlated with the PASI score (r = 0.44, *p* = 0.002) [55]. Basavaraj et al. also demonstrated significant differences between the levels of 8-hydroxy 2′-deoxyguanosine and the different psoriasis severity stages (mild, moderate, severely evaluated by the PASI score) (*p* < 0.05) [53]. Regarding plasma nitric oxide levels, there was an association with the subjects’ age in the exploration of Sikar Akturk et al. (r = −0.43, *p* = 0.04) [52]. An assessment by Gabr and Al-Ghadir detected notable negative association of malondialdehyde and nitric oxide, as well as positive correlations of total antioxidant status, superoxide dismutase and catalase (*p* < 0.001) with different forms of psoriasis (mild, moderate, severe) [47]. In addition, malondialdehyde, nitric oxide, total oxidant status, superoxide dismutase, catalase alterations were observed by Kadam et al. in their study of a population with psoriasis from India. Both investigations suggested a decrease in antioxidant systems due to overexpression of inflammatory mechanisms, leading to an increase in lipid peroxidation, thus maintaining this cycle. In terms of nitric oxide, levels, research on 58 subjects from Saudi Arabia found higher levels in patients with severe versus moderate/mild forms (*p* < 0.001). Thus, a positive correlation was observed between nitric oxide levels and PASI (r = 0.51, *p* < 0.001), but not between nitric oxide levels and disease duration [42]. In addition to malondialdehyde and catalase whose correlation with disease severity was highlighted in numerous studies, Pujari et al. highlighted a correlation between vitamin E levels (as a marker of oxidative stress) and the severity of the disease expressed by the PASI score [40].

Another marker of oxidative stress that correlates with the severity of the disease expressed by the PASI score is ischemia-modified albumin. Chandrashekar et al. detected a positive correlation not only between ischemia-modified albumin and PASI (r = 0.71, *p* < 0.001), but also between CRP and PASI (r = 0.89, *p* < 0.001). The same study showed a negative correlation between the level of 25-OH-vitamin D and PASI (r = −0.71, *p* < 0.001), CRP and ischemia-modified albumin, emphasizing once again the role that vitamin D has in modulating inflammation and in decreasing oxidative stress levels [38]. The concentrations of methylglycoxal correlate with psoriasis severity (r = 0.58, *p* = 0.008) and are dependent at the same time on total antioxidant status (*p* < 0.001), total peroxide concentration and oxidative stress index (*p* < 0.001) levels [44]. Another marker recently investigated in terms of cellular stress in psoriasis is hemoxygenase. This is an enzyme that degrades heme, being extremely important in the defense against cellular stress and immunomodulation, and has compensatory high values in individuals with this dermatosis, correlating both with weight (r = 0.29, *p* = 0.03) and PASI (r = −0.62, *p* < 0.001) [26]. Moreover, the total level of plasma thiols is negatively associated with age (r = −0.479, *p* < 0.0001), disease duration (r = −0.217, *p* = 0.04) and PASI (r = −0.217, *p* = 0.04) [28]. 

However, many studies did not find significant correlations between oxidative stress parameters and the severity or the duration of psoriasis. For example, Kirmit et al. evaluated the associations of myeloperoxidase, ischemia modified albumin, catalase and CRP with the disease severity based on the affected area and the only correlation found was between CRP and the mild/moderate form of the disease (*p* = 0.04) [14]. Furthermore, Ozemir et al. detected no correlation of ischemia-modified albumin with PASI, BMI, disease duration or age [48]. Neither did Kizilyel et al. find any associations of total oxidant status, total antioxidant status, malondialdehyde, 8-hydroxy 2′-deoxyguanosine levels and disease severity [19,43]. Moreover, although in several studies conducted in Egypt and India correlations were observed between the levels of total antioxidant status, malondialdehyde, superoxide dismutase and psoriasis severity, in the research of Baz et al. who analyzed the Turkish population, these correlations were not confirmed [64]. What is more, in terms of antioxidant balance, although it is altered in individuals diagnosed with psoriasis, no correlations were identified between superoxide dismutase, glutathione peroxidase, catalase and age, PASI score or disease duration in Karaman et al. ’s investigation [61]. Neither did Hashemi et al. detect any correlation between adenosine deaminase, serum trypsin inhibitory capacity or total antioxidant status and the severity nor duration of the disease, despite statistically significantly differences in psoriasis subjects versus controls [56]. Similar results were obtained by Kaur et al. who reported that in psoriasis oxidative stress markers do not correlate with age, disease duration or severity, despite the fact that there are notable alterations of the redox balance, i.e., increased oxidative markers and decreased antioxidant systems [44,53].

Similarly to Kizilyel et al. ’s research, no significant correlations were noted between malondialdehyde and PASI in the study of Skoie et al. but there was a pronounced increase in advanced oxidation protein products in patients with PASI > 7 (*p* = 0.05) [18]. Haberka et al. did not detect any differences in terms of advanced oxidation protein products, visfatin and nesfatin levels and the severity or duration of the illness [24]. Neither did Emre et al. find any correlations between paraoxonase 1/arylesterase concentration levels and the disease status [28]. No correlations were recorded between plasma and tissue nitric oxide and malondialdehyde levels in terms of disease severity, disease duration, or patient sex [52]. Although their levels were elevated, paraoxonase 1 and arylesterase were not associated with the number of metabolic comorbidities in individuals with psoriasis and metabolic syndrome [50]. The same observations were highlighted by Elaine Husni et al. who concluded that, in spite of lower levels of paraoxonase 1and arylesterase in patients with Ps and PsA, the concentrations of these biomarkers are not related to the presence of cardiovascular comorbidities [23].

Myeloperoxidase, protein carbonyl compounds, advanced oxidation protein products, lipid hydroperoxides and pyrrolized proteins, although they are significantly increased in patients with psoriasis, regardless of PASI, and do not correlate with the severity of the disease, there are no significant differences between groups of patients with mild, moderate or severe forms [34]. Another study by Relhan et al. shows that although the plasma levels of thiols and malondialdehyde differ significantly between acute and chronic forms of the disease, an association with PASI was observed only in the plasma level of malondialdehyde in patients in remission, with PASI over six compared with those with PASI under six (*p* < 0.05) [66]. The levels of trace elements and acute phase proteins (Cu, Fe, transferrin, ceruloplasmin), although significantly altered in patients with psoriasis, do not seem to correlate with the severity or duration of the disease [66].

### 3.2. Polymorphisms of Genes Encoding Markers or Enzymes of Oxidative Stress in Psoriasis

A total of eight studies evaluated polymorphisms in genes encoding markers or enzymes of oxidative stress, such as: glutathione S-transferase/gluyathione, paraoxonase 1, malondialdehyde, arylesterase, apolipoprotein B (APOB), apolipoprotein A1 (APOA1), methylentetrahydrofolatereductase (MTHFR), vascular adhesion protein-1 (VAP-1), superoxide dismutase 1, superoxide dismutase 2, nuclear factor erythroid 2 like 2 (NFE2L2), cytochrome b-245 beta chain (CYBB), interleukin-17 (IL-17), lipoprotein (a) [LP(a)], tumor necrosis factor-α (TNF-α), interleukin-10 (IL-10), N-acetyltransferase (N-AT), endothelial nitric oxide synthase (eNOS), cationic amino acid transporters (CATs), receptor for advanced glycation-end products (RAGE) and glutathione S-transferase [67,68,69,70,71,72,73,74]. Table 2 summarizes the information presented in the aforementioned studies.

Guarneri et al. assessed the frequency of glutathione S-transferase M1/glutathione S-transferase T1(GSTM1/GSTT1) polymorphisms in 148 individuals with psoriasis versus 148 age-matched healthy counterparts, revealing that both glutathione S-transferase T1 null and GSTM1/GSTT1 “double null” genotypes are associated with psoriasis. Moreover, the association between this ailment and glutathione S-transferase T1 null was stronger in women versus men [68].

Asefi et al. examined the paraoxonase-1 55 polymorphism in a group of 200 psoriasis patients versus healthy controls and discovered that the paraoxonase-1 55 M allele is a risk factor for psoriasis. Moreover, paraoxonase-1 55 M allele carriers displayed higher malondialdehyde (*p* < 0.001), apolipoprotein B/apolipoprotein A1 ratio (*p* = 0.004), apolipoprotein B (*p* = 0.001) and lipoprotein (a) (LP (a)) (*p* = 0.03), but lower arylesterase activity (*p* = 0.03) [72]. Furthermore, in terms of paraoxonase-1 gene polymorphisms, it was observed that the presence of the rs662 polymorphism (A > G) is more frequently associated with the development of psoriasis (*p* = 0.004), with affected cases being most likely carriers of the heterozygous form A/G (*p* = 0.003). In addition, the G rs662 allele (A > G) was more commonly encountered in individuals suffering from this disorder (*p* = 0.003), emphasizing the role this allele could play in increasing one’s susceptibility of psoriasis [68].

Another research study published by Asefi et al. explored the contribution of another polymorphism in a gene responsible for the elevation of oxidative stress markers, namely malondialdehyde, vascular adhesion protein-1 and apolipoprotein B, to the risk of developing psoriasis. Thus, subjects with the methylentetrahydrofolatereductase -677-T (C/CC + CT/TT) and the T allele of the methylentetrahydrofolatereductase -677 gene not only have an up to 7.5 times higher risk of developing psoriasis but display also an elevated risk of cardiovascular complications [71]. Another gene whose polymorphisms appear to be involved in psoriasis is inducible nitric oxide synthase. Chang et al. noted that there is a significant association between pentanucleotide repeatability (CCTTT) in the inducible nitric oxide synthase gene promoter region and psoriasis susceptibility of the Chinese population from Taiwan. Inducible nitric oxide synthase promoter activity increases in parallel with the repeat number of (CCTTT). In addition, the LL-type genotypes (with a sequence repeatability greater than 14) are associated with a lower risk of developing psoriasis in the general population. Individuals who exhibit this genotype are less likely to experience the onset of psoriasis under the age of 40 (*p* = 0.04) [70].

A key molecule in psoriasis is catalase. Cationic amino acid transporters are transporters responsible for ensuring adequate levels of L-arginine in the body which is an amino acid that serves as substrate for two relevant enzymes for the integument, namely inducible nitric oxide synthase and arginase that are involved in the regulation of keratinocyte proliferation. Schnorr et al. demonstrated that in the skin of patients with psoriasis, both those affected by psoriatic plaques and those apparently spared, there is an overexpression of the catalase 1 isoform of this transporter. Only this isoform is notably elevated in psoriasis versus healthy controls (*p* < 0.05), whereas the distribution of the other two isoforms, i.e., catalase 2A and catalase 2B, is not. This particular isoform can potentially limit the activity of the inducible nitric oxide synthase gene [73]. Receptors for advanced glycation-end products polymorphisms have also been described in this dermatological disorder, with an association of the 2184A/G polymorphism and of the AA homozygous form with the onset of plaque psoriasis (*p* = 0.001). However, no significant correlations were detected between the frequency of these polymorphisms and the presence of early onset family forms under the age of 40 years [69].

### 3.3. The Effect of Anti-Psoriasis Therapy on Oxidative Stress Markers

A total of 15 studies evaluated the impact of different standard therapies for psoriasis, i.e., methotrexate (MTX), narrowband ultraviolet B (NB-UVB), Goeckerman therapy (GT) (combined exposure of 3% crude coal tar ointment and UV radiation), broadband ultraviolet B (BB-UVB), psoralen and UVA therapy (PUVA) and biologic therapy, on severity of psoriasis, as evaluated by PASI or Dermatology Life Quality Index (DLQI), as well as on several parameters of oxidative stress or other biochemical markers: ascorbyl radicals (Asc), total antioxidant capacity (TAC), protein carbonyl content (PCO), gluthatione (GSH,) CRP, thiobarbituric acid reactive substances (TBARS), total oxidative status, oxidative stress index, arylesterase (AS), total cholesterol (TC), low density lipoprotein (LDL), high density lipoprotein (HDL), alanine aminotransferase (ALT), thyroid-stimulating hormone (TSH), fumaric acid esters (FAE), Cu, selenium (Se), zinc (Zn), 4-hydroxy nonenal-protein (4-HNE), polyunsaturated fatty acids (PUFA), benzo[a]pyrene-7,8-diol-9,10-epoxide (BPDE), GPx, NG-monomethyl-l-arginine (1-NMMA), total bilirubin (TB), membrane-bound hemoglobin (MBH), membrane protein band 3 (MPB3), lecithin-cholesterol acyltransferase (LCAT), serum amyloid A (SA-A), cytochrome p450 (CYP 450), 8-hydroxy-2′-deoxyguanosine (8H2DG), 8-hydroxyguanosine (8HG) and 8-hydroxyguanine (8HGN) [75,76,77,78,79,80,81,82,83,84,85,86,87,88,89]. Table 3 summarizes the information presented in the aforementioned studies.

Elango et al. studied the impact of 7.5 mg/week methotrexate treatment for 12 weeks in patients with psoriasis by assessing various parameters of oxidative stress in the serum and evaluated via biopsy of psoriasis lesions and lesion-free integument. The authors ob-served that methotrexate therapy resulted in a significant increase in reactive oxygen species, plasma malondialdehyde and the expression of caspase-3 (*p* < 0.001). However, although superoxide dismutase, catalase and total antioxidant status displayed a decreasing trend, there were no significant pre-/post-methotrexate administration differences [76]. Tekin et al. also reported that 20 mg/week methotrexate notably reduced serum nitrite–nitrate levels in 22 individuals suffering from psoriasis (*p* < 0.05). However, no correlations were detected between nitrite–nitrate concentrations pre/post-methotrexate administration and the PASI score or between post-treatment levels and the dose of methotrexate received [78]. Akbulak et al. evaluated the effects of 10–15 mg/week methotrexate for ≥12 weeks on the expression of GST and CYP isoenzymes in 21 psoriasis subjects. There were no alterations of the expression of Glutathione S-transferases-K1, Glutathione S-transferases-K1, Glutathione S-transferases-T1, cytochrome p450-1B1, and cytochrome p450-2E1 pre–post methotrexate treatment, however their expression was elevated post-methotrexate administration in the psoriasis versus control group. At the same time, Glutathione S-transferase-O1 expression was comparable in patients with psoriasis versus healthy counterparts [76]. To assess whether the pro-oxidant effect of methotrexate is involved in its systemic side effects, Kılınc et al. evaluated the lipid profile and several oxidative stress indices in 26 psoriasis patients. Systemic treatment with methotrexate for weeks alleviated dyslipidemia, i.e., a notable reduction in total cholesterol, LDL and HDL was seen (*p* < 0.05), however it did not influence oxidative stress levels, as similar pre- versus post-therapy concentrations of serum paraoxonase-1, total antioxidant status, total oxidant status, and oxidative stress index were registered (*p* > 0.05). Moreover, there were no significant correlations of the oxidant status with serum lipids or the PASI score (*p* > 0.05), despite a significant decrease in post-treatment PASI (*p* < 0.001) [77]. Darlenski et al. analyzed a group of 22 patients with psoriasis and investigated the evolution of several indices after performing 14 sessions of NB-UVB 311 nm therapy. At the end of the phototherapy sessions, both PASI and DLQI scores improved significantly (*p* < 0.001). In addition, there was a significant reduction in trans–epidermal water loss, stratum corneum hydration, and oxidative stress parameters, namely reactive oxygen species, Asc and malondialdehyde (*p* < 0.001). These findings emphasize the systemic role of phototherapy in alleviating psoriasis and its impact on the modulation of oxidative stress in the integument, as the skin represents an important source of free radicals that act not only at their production site but can exhibit systemic actions in the human body [80]. Similarly, Karadag el al. employed NB-UVB 311 nm phototherapy in the management of this dermatological disorder and investigated its influence on the expression of several enzymes. It was observed that the expression of Glutathione S-transferases-K1, Glutathione S-transferases-K, Glutathione S-transferases-T1, cytochrome p450-1A1 and cytochrome p450-2E1 was significantly higher in individuals with psoriasis versus the control group (*p* = 0.02, *p* = 0.001, *p* = 0.006, *p* = 0.003 and *p* = 0.001). However, no significant post- versus pre-treatment differences were exhibited by the 32 psoriasis subjects for any of the aforementioned enzymes [82]. Similarly, Pektas et al. explored the actions of NB-UVB 310-315 nm phototherapy on the oxidative status of 24 subjects with psoriasis; total oxidant status and oxidative stress index values increased significantly (*p* < 0.001) after the phototherapy sessions but, at the same time, hsCRP, total antioxidant status, serum paraoxonase-1 and AS displayed similar values pre-/post-exposure (*p* > 0.05). Moreover, no correlation was noted between PASI and the aforementioned parameters of the oxidative index [83]. Furthermore, Karaarslan et al. analyzed the systemic effect of BB-UVB by measuring thiobarbituric acid reactive substance and nitrite–nitrate levels in 32 psoriasis patients. No statistically significant changes were exhibited in the study versus the control group pre-phototherapy in terms of thiobarbituric acid reactive substance and nitrite–nitrate concentrations. Thiobarbituric acid reactive substance and nitrite–nitrate levels increased significantly after BB-UVB treatment (*p* < 0.05 and *p* < 0.01, respectively). No associations were detected between the duration of the disease, its se-verity or the total dose of UVB therapy administered and the serum levels of thiobarbituric acid reactive substance and nitrite–nitrates. However, there was a negative correlation between the total nitrite and thiobarbituric acid reactive substance levels (r = −0.58, *p* = 0.03) [85]. A similar study was conducted by Darlenski et al. who followed the dynamics of serum carotenoid levels after 10 sessions of NB-UVB phototherapy. This psoriasis management strategy induced clinical improvements of the skin disorder, as quantified by the PASI and DLQI scores (*p* < 0.001). Even though the serum levels of carotenoids in the study group were lower than in the controls (*p* < 0.001), NB-UVB therapy resulted in a statistically non-significant reduction in psoriasis subjects [79]. Wacewicz et al. recruited 118 individuals with psoriasis and healthy controls who were treated with NB-UVB phototherapy to find out the importance of various minerals, e.g., Se, Zn or Cu, as well as of CRP and total antioxidant status, in the pathogenesis of this skin disease. They registered that NB-UVB determined decreases in Se and total antioxidant status values (*p* < 0.05), whereas Cu, Zn, and the Cu/Zn ratio remained unchanged [81]. Coimbra et al. explored, in their longitudinal cross-sectional research, erythroid disturbances in 43 psoriasis subjects managed with NB-UVB, PUVA and calcipotriol. According to their results, patients with psoriasis exhibit significantly higher leukocytes, neutrophils, elastase, lactoferrin, CRP and thiobarbituric acid values, and also an increase in total antioxidant status and thiobarbituric acid/total antioxidant status ratio, but without statistical significance for the last ones. Moreover, at the end of the 12 weeks of treatment, a significant decrease in leukocytes, neutrophils, elastase, lactoferrin, CRP, thiobarbituric acid and thiobarbituric acid/total antioxidant status was observed. There were also significant alterations of the psoriasis area and severity index score in individuals managed with topical products and with PUVA (*p* = 0.01) but not with NB-UVB (*p* > 0.05) [84]. Barygina et al. detected an alleviation of oxidative stress in psoriasis subjects who received 5 mg/kg of infliximab every 8 weeks for 6 months. Although malondialdehyde, protein carbonyl content, reactive oxygen species, lipoperoxidation and GC levels were elevated in this affliction of the integument, protein carbonyl content, thiobarbituric acid reactive substance, thiobarbituric acid reactive substance and reactive oxygen species decreased significantly (*p* < 0.05), whereas total antioxidant status increased (*p* < 0.05), following infliximab administration [78]. Another anti-TNF-α monoclonal antibody used to treat psoriasis is efalizumab. As no marker has yet been agreed to monitor the effectiveness of this treatment, Pastore et al. measured a number of clinical and laboratory indices in 26 psoriasis patients. This drug elevated the levels of IL-8 and vascular endothelial growth factor yet did not alter the levels of IL-10, IFN-γ or the content of polyunsaturated fatty acids membranes. However, it decreased nitrites/nitrates, catalase and malondialdehyde levels and stimulated the GPx and glutathione S-transferases activity [88]. The systemic actions of two other biological psoriasis management options, namely adalimumab and etanercept, were investigated Campanati et al. At the end of 12 weeks of therapy with one of these medications, it was observed that vascular endothelial growth factor production decreased significantly (*p* < 0.05), with adalimumab exerting a more notable effect in reducing this variable. Both agents caused an increase in superoxide dismutase and glutathione content and a decrease in the nitric oxide production by inducible nitric oxide synthase, catalase and glutathione S-transferases, with variable changes depending on the skin type evaluated: intact skin, perilesional or lesional [89]. Dyslipidemia is often evidenced is individuals suffering from psoriasis. Wolk et al. verified the changes in serum lipids and oxidative stress markers induced by tofacitinib, an oral Janus kinase inhibitor. Although this pharmacological agent significantly increased LDL-C and HDL-C (*p* < 0.05), the total cholesterol/HDL-C ratio remained constant. Moreover, the activity of paraoxonase-1 increased (*p* < 0.05), whereas inflammation indices, e.g., serum amyloid A and CRP, decreased (*p* < 0.05) [87].

## 4. Discussion

This systematic review investigated the involvement of oxidative stress plays in the pathophysiology of psoriasis by the qualitative synthesis of information derived from 79 original research papers that evaluated the levels of a wide variety of oxidative stress markers, e.g., total antioxidant status, total oxidant status, malondialdehyde, catalase, myeloperoxidase, ischemia modified albumin, advanced oxidation protein products, NADPH oxidase, reactive oxygen species, superoxide dismutase, lipid hydroperoxides, and others in patients diagnosed with this skin disorder, as well as the influence of several polymorphisms in genes which play relevant roles in the redox balance. Moreover, we analyzed the impact of anti-psoriasis therapy, i.e., UVB phototherapy, methotrexate, infliximab, adalimumab, etanercept or tofacitinib, on several indices of oxidative stress.

According to the data in the literature, oxidative stress plays an important role in initiating and perpetuating chronic diseases, e.g., cardiovascular, liver, neurological, metabolic, endocrinological and dermatological disorders [90]. Of the latter, psoriasis is an important representative, affecting a high percentage of the general population. The way in which these parameters reflect the level of oxidative stress in the body is difficult to assess, as so far there are no clear correlations between the oxidative balance and various oxidizing enzymes, antioxidant molecules or oxidation products, as the latter are generated in the body via multiple mechanisms including in physiological conditions [91,92]. The main source of reactive oxygen species remains the mitochondria, namely the mitochondrial inner membrane. Moreover, reactive oxygen species generation also results in the alteration of the main cellular components: lipids, carbohydrates and proteins [91,92]. Thus, at the lipid level, an increased number of reactive oxygen species is responsible for the enzymatic and non-enzymatic peroxidation of polyunsaturated fatty acids and LDL (with the formation of oxidized LDL), affecting the cell membrane and eventually leading to apoptosis [93]. Regarding the action of reactive oxygen species on the protein and carbohydrate components, their presence is associated with the formation of carbonyl-type compounds, advanced glycation end-products and advanced oxidation protein products, with an additional role of stimulating reactive oxygen species-generating processes at the cellular level. In addition to mitochondrial free radicals, a large number of reactive species also result from the action of pro-oxidant enzymes, such as xanthine oxidase, nitric oxide synthase, myeloperoxidase and NADPH oxidase. The body’s defense mechanisms against the increase in ROS levels above the physiological levels consist of enzymatic antioxidant systems (superoxide dismutase, catalase, glutathione peroxidase, etc.) which act as scavengers that capture the already formed free radicals, but also non-enzymatic antioxidant systems (glutathione, antioxidant vitamins) that have the role of interrupting reactive species-generating reactions [93]. Thus, glutathione, one of the main antioxidant systems, has the role of decreasing the reactive oxygen species concentrations via donation of a hydrogen ion with a neutralizing role. Thus, oxidized glutathione is generated [94]. Moreover, the increase in reactive oxygen species also causes changes at the nuclear level, by activating the transcription factors Nrf2 (nuclear factor erythroid related factor 2) and NF-kB (nuclear factor kappa light chain enhancer of activated B cells) which further contribute to the anti-oxidative stress defense mechanisms by inducing the synthesis of antioxidant molecules [93].

Figure 2 depicts role of oxidative stress in the initiation and evolution of psoriasis and its associated comorbidities.

Thus, following the analysis of the included studies, most papers delineated a significant alteration of the redox balance, with a significant decrease in antioxidant enzymes and antioxidant markers and an increase in pro-oxidant molecules in psoriasis. A conundrum of research endeavors which collected venous blood samples from individuals suffering from psoriasis with variable duration of the disease detected low levels of total antioxidant status [37,38,46,47,51,54,63,64] and alterations in enzymes involved in decreasing free oxygen radicals concentrations, i.e., catalase [21,22,39,40,47,54,61], superoxide dismutase [39,47,54,61,63,65], paraoxonase-1 [15,23,39,43,50,51] and glutathione peroxidase [25,63]. Several studies with contradictory results were also present, highlighting that in psoriasis a significant elevation in the levels of the aforementioned antioxidant molecules can also occur [14,25,61,65]. We may hypothesize that, in psoriasis, there is a compensatory increase in antioxidant systems in order to counterbalance the elevated levels of oxidative stress. The plasma concentrations of pro-oxidant molecules was notably increased in all studies, reinforcing the theory that oxidative stress is a key player in the pathogenesis of this disease. The main indices assessed were total oxidative status, reactive oxygen species, myeloperoxidase, ischemia modified albumin, advanced glycation end-products and advanced oxidation protein products whose values were higher in subjects with psoriasis versus their healthy counterparts [14,16,19,20,21,22,24,29,33,37,45,46,48]. Moreover, some of these markers were correlated with the duration and severity of the disease, as well as indices of atherosclerosis. Consequently, these data support the contribution of oxidative stress to the development and evolution of the aforementioned illnesses, as well as to the crosstalk of psoriasis and several cardiometabolic comorbidities, i.e., atherosclerosis, hypertension or obesity, with which it is commonly associated [20,23,27,32,36]. In addition, several gene polymorphisms in genes encoding molecules with a role in the redox balance, such as glutathione S-transferase M1/glutathione S-transferase T1, paraoxonase-1, methylentetrahydrofolatereductase, glutathione S-transferase and catalase were upregulated or more frequently expressed in psoriasis [67,68,69,70,71,72,73,74]. For example, an enzyme with an important role in the psoriasis-associated comorbidities is paraoxonase-1. Paraoxonase-1 plays an antioxidant role, both by protecting LDL against peroxidation and by its direct role in eliminating oxidized forms of fatty acids, thus having both an antioxidant and an antiatherogenic action. Psoriasis is associated with the presence of paraoxonase-1gene polymorphisms that cause a low paraoxonase-1 activity, as well as an increase in oxidative stress [68]. Moreover, certain polymorphisms (paraoxonase-1 55 M > L) appear to be factors that directly impact on one’s risk of developing psoriasis [72].

On one hand, the use of anti-psoriasis therapy displayed promising results in reducing oxidative stress levels. On the other hand, the interpretation of the findings of studies exploring this research topic must be conducted with caution. Only a limited number of papers was published on this subject, involving a small number of cases but a wide range of management options, varying from systemic therapy, i.e., the immunosuppressive agent methotrexate and monoclonal antibodies, to local approaches, e.g., phototherapy. Thus, although biological agents and phototherapy seem to be linked with a decrease in reactive oxygen species and malondialdehyde and an increase in total antioxidant status, these results were not confirmed by all investigated papers. Of these, some highlighted a reduction in total antioxidant status and a paralleled elevation in total oxidative status despite disease amelioration [77,81,82,83,88]. Moreover, methotrexate has been proven to display pro-oxidant effects and it is known that this pharmacological agent enhances oxidative stress levels, mainly via post-administration generation of reactive oxygen species and malondialdehyde [76]. However, methotrexate has a dual effect on oxidative stress levels. On one hand, it enhances the generation of reactive oxygen species and, on the other hand, it inhibits the synthesis of nitric oxide. The administration of methotrexate is, thus, associated with a decrease in nitric oxide concentrations [76]. Thus, the antiproliferative and immunosuppressive effects of this pharmacological agent in psoriasis are mainly determined by the aforementioned increase in reactive oxygen species that induces the apoptosis of keratinocytes [76,77]. This pro-oxidant effect is also responsible for some of the side effects of this therapy which seem to be alleviated by the administration of antioxidant compounds [77]. Another molecule with antioxidant and protective roles is paraoxonase-1 which in patients with psoriasis treated with methotrexate is present in low concentrations, possibly due to the toxic effect of methotrexate on the liver which is the synthesis site of paraoxonase-1 [77].

Other pharmacological agents commonly used in the treatment of psoriasis are TNF-alpha inhibitors, namely infliximab. The use of this drug is associated with a decrease in oxidative stress levels due to its property to block the pro-oxidant role of TNF-alpha. Moreover, as opposed to the use of methotrexate which is linked with an elevation in pro-oxidant levels, infliximab does not influence the pro-oxidant/antioxidant balance or can influence it in the favor of antioxidant molecules [86].

Our paper has a number of strengths and limitations. One of the main strengths of our manuscript is the exhaustive analysis of the scientific literature. Our qualitative synthesis included 79 studies that assessed markers of oxidative stress and their contribution to the development and evolution of psoriasis and also highlighted their associations with the severity of the disease or the frequency of certain complications. Moreover, we also explored the impact of psoriasis management options on several of these oxidative stress indices. To our view, this an important aspect to take into consideration, peculiarly due to the crosstalk of psoriasis and cardiometabolic complications/comorbidities. However, despite the large number of articles included in our research and the large amount of information extracted, the significant heterogeneity of the measured parameters, the different assessment techniques employed, and the variety of samples collected did not allow us to quantitatively explore the association of oxidative stress and psoriasis by means of a meta-analysis. However, to our knowledge, this is the first systematic review to evaluate such a large number of indices of oxidative stress and it highlights the role that this complex phenomenon can play in the pathophysiology of this disease. Further research, preferably cohort studies, is needed in the future to delineate the accuracy of novel serum/plasma, urine or saliva oxidative stress biomarkers, such as ischemia modified albumin, biopyrins, 8-hydroxy guanosine, advanced oxidation protein products and others.

## 5. Conclusions

Oxidative stress seems to be involvement in the development and evolution of psoriasis and its associated comorbidities, in particular those affecting cardiometabolic health. The utility of the assessment of circulating serum, plasma, urinary and/or skin biomarkers of oxidative stress and of the study of polymorphisms in genes regulating the redox balance remains to be explored in future prospective cohort studies. However, the role that oxidative stress plays in the pathophysiology of psoriasis is indisputable and understanding the mechanisms by which it contributes to the initiation and maintenance of this chronic condition can aid the management of this dermatosis in the near future. Thus, such markers may emerge as essential tools in the early diagnosis of psoriasis, from the subclinical stage, as well as in evaluating the pattern of evolution and the therapeutic response.

## Figures and Tables

**Figure 1 antioxidants-11-00282-f001:**
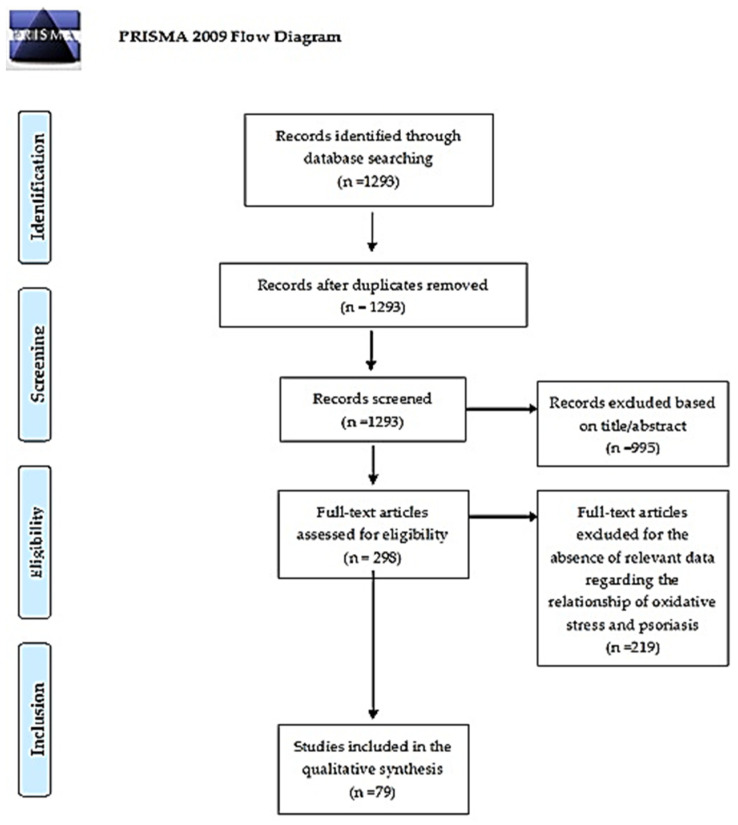
PRISMA 2009 Flow Diagram. From Moher, D.; Liberati, A.; Tetzlaff, J.; Altman, D.G. The Prisma Group. Preferred reporting items for systematic reviews and meta-analyses: The PRISMA statement. PLoS Med. 2009, 6, e1000097, doi:10.1371/journal.pmed.1000097. For more information, visit www.prisma-statement.org (accessed on 9 January 2022) [11].

**Figure 2 antioxidants-11-00282-f002:**
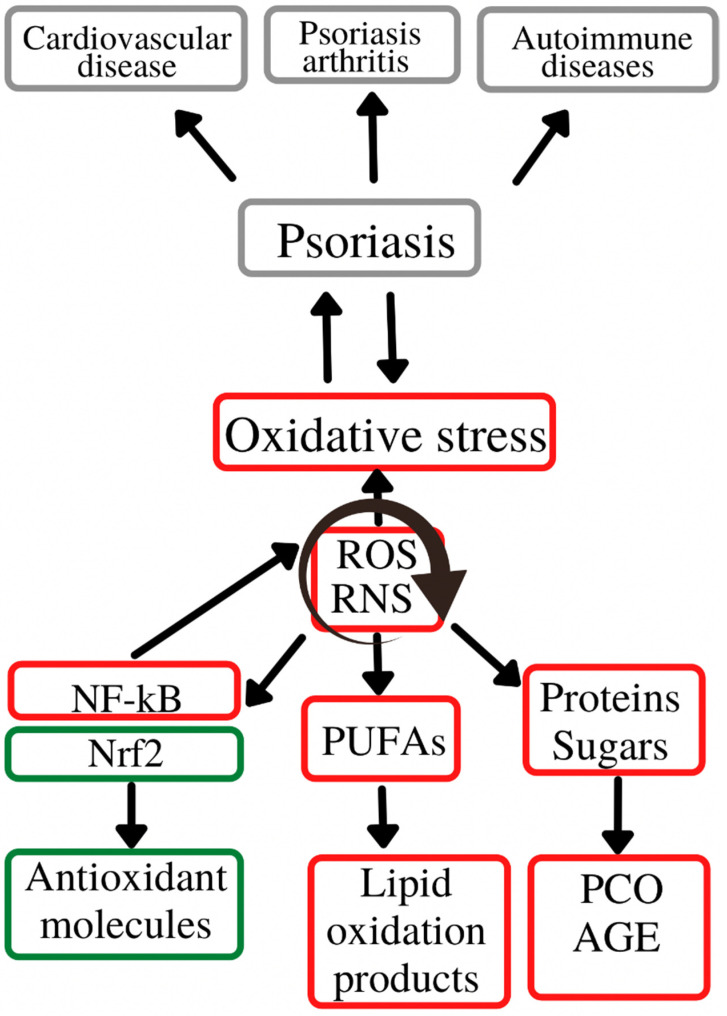
Oxidative stress in the psoriasis evolution. Oxidative stress plays an important role in the initiation and evolution of psoriasis and its associated-comorbidities/complications (cardiovascular diseases, psoriasis arthritis and autoimmune diseases. These complications are generated by the chronic action of ROS and RNS on the main cell constituents (lipids, proteins and carbohydrates), with a secondary activation of molecular pathways (Nrf2, NF-kB) responsible for inducing synthesis of antioxidant molecules (Nrf2) or pro-oxidant enzymes (NF-kB). ROS—reactive oxygen species; RNS—reactive nitrogen species; Nrf2—nuclear erythroid factor 2-related factor 2; NF-kB—nuclear factor kappa light chain enhancer of activated B cells; PUFAs—polyunsaturated fatty acids; AGE—advanced glycation end-products; PCO—protein carbonyl content.

**Table 1 antioxidants-11-00282-t001:** Markers of Oxidative Stress in Patients with Psoriasis.

	Authors and Year	Country	Type of Study	Sample Size (Psoriasis/Controls)	Mean Disease Duration (Years)	Mean Age (Years)	Sample	Measured Parameters	Main Results (Psoriasis Versus Control Group)
1.	Kirmit et al. 2020 [14]	Turkey	Case–control study	147 (87/60)	7 (0–37)	32.8 ± 15.6	Venous blood	CRP, CAT, MPO, FOX, IMA	↑CAT, FOX, IMA (*p* < 0.001), CRP (*p* = 0.04) ↑MPO (*p* = 0.123)
2.	Oszukawska et al. 2020 [15]	Poland	Case–control study	96 (66/30)		21–73 years	Venous blood	PON-1, alpha-tocopherol, uric acid, homocysteine	↑Uric acid, homocysteine ↓PON-1 (*p* < 0.001), alpha tocopherol (*p* < 0.05)
3.	Skutnik-Radziszewska et al. 2020 [16]	Poland	Case–control study	80 (40/40)	12.7 ± 9	45.6 ± 20.2	Venous blood, stimulated/unstimulated saliva	Px, CAT, SOD (saliva), TOS, OSI, AGE, AOPP, MDA, LOOH (blood and saliva)	↑TOS, OSI, AGE, AOPP, MDA, LOOH (*p* < 0.001) (venous blood and saliva) ↑ROS (*p* < 0.001) (saliva)
4.	Kiafar et al. 2020 [17]	Iran	Observational study	20 (20/0)	-	38.9 ± 12.6	Skin biopsy samples	TrxR	↓TrxR (*p* < 0.01)
5.	Skoie et al. 2019 [18]	Norway	Case–control study	168 (84/84)	14 (8–24)	45	Venous blood	AOPP, MDA, CRP	AOPP unmodified ↓MDA (*p* = 0.03)
6.	Kizilyel et al. 2019 [19]	Turkey	Case–control study	95 (50/45)	8.8 ± 6.9	32.5 ± 14.5	Venous blood	TOS, TAS, MDA, 8H2D	↑TOS (*p* < 0.001)
7.	Ergun et al. 2019 [20]	Turkey	Case–control stody	72 (50/20)	-	-	None (non-invasive measurements)	AGE	↑AGE (*p* = 0.05)
8.	Wojcik et al. 2019 [21]	Poland	Case–control study	48 (32/16)	-	35 (Ps) 37(PsA)	Venous blood	NADPH oxidase, Xanthine oxidase, ROS, CAT, GSH-Px, GSH	↑NADPH, xanthine oxidase, ROS (PsA > Ps) (*p* < 0.05) ↓CAT, GSH (*p* < 0.05)
9.	Esmaeili et al. 2019 [22]	Iran	Case–control study	20 (10/10)	18.5 ± 3.1 (mild Ps), 13.5 ± 7.8 (moderate/severe Ps)	37 (mild Ps) 27 (moderate/severe Ps)	Venous blood	GSH, ROS, TAS, FRAP	↑ROS (*p* = 0.04) ↓CAT (*p* = 0.02)
10.	Elaine Husni et al. 2018 [23]	USA	Cross sectional study	688 (343 (198PsA, 143 Ps)/345)	15.5 ± 13.2 (Ps), 21.1 ± 14.9 (PsA)	45.7 ± 15.3 (Ps), 50.4 ± 11.8 (PsA)	Venous blood	PON-1, AS	↓AS (*p* < 0.001), similar PON-1 levels
11.	Haberka et al. 2018 [24]	Poland	Case–control study	119 (80/39)	15.3 ± 11.2	43 ± 13.5	Venous blood	Visfatin, Nesfatin, AOPP	↑AOPP, visfatin
12.	Ambrozewicz et al. 2018 [25]	Poland	Case–control study	102 (68/34)	-	38.2	Venous blood	NADPH oxidase, Xanthine oxidase, GSH-Px, GSH-R, SOD, TrxR, GSH, Vitamin C	↑NADPH oxidase, xanthine oxidase (Ps and PsA), SOD (only in Ps) (*p* < 0.05) ↓Trx, TrxR, GSH, Vitamin C (Ps and PsA), GSH-Px (only in PsA) (*p* < 0.05)
13.	El-Rifaie et al. 2018 [26]	Egypt	Case–control study	101 (51/50)	-	45.6 ± 15.1	Venous blood	HO	↑HO (*p* < 0.001)
14.	Asha et al. 2017 [27]	India	Case–control study	300 (150/150)	-	39.6 ± 11.9	Venous blood	OxLDL, GSH, FRAP, MDA	↑OxLDL, OxLDL/LDL (*p* < 0.01), MDA(*p* < 0.001) ↓GSH
15.	Emre et al. 2017 [28]	Turkey	Case–control study	166 (90/76)	-	36.0	Venous blood	Native SH, Total SH, SS	↑Native SH (*p* = 0.013), Total SH (*p* = 0.04)
16.	Papagrigoraki et al. 2017 [29]	Italy	Cross sectional study	160 (120 (80 Ps, 40 eczema)/40)	-	48 ± 8 (severe Ps). 47 ± 11 (mild Ps)	Venous blood	AGE(s) AGE(*p*)	↑AGE(s) (*p* = 0.01) AGE(*p*) (*p* = 0.01)
17.	Shahidi-Dadras et al. 2017 [30]	Iran	Case–control study	80 (40/40)	10.1 ± 8.3	36.7 ± 14.8	Venous blood	Cu, Fe, Trf, Cp	↓Fe, Trf (*p* < 0.01) ↑Cp (*p* = 0.02)
18.	Bakry et al. 2016 [31]	India	Case–control study	115 (85/30)	0.3 ± 0.2	39.8 ± 18.1	Urine	Urinary biopyrrins	↑Urinary biopyrrins (*p* < 0.001)
19.	Sunitha et al. 2016 [32]	India	Cross sectional study	90 (45/45)	3.7 ± 5.1	44.9 ± 14.3	Venous blood	AuAb-oxLDL, oxLDL	↑AuAb-oxLDL, OxLDL (*p* < 0.001)
20.	Dilek et al. 2016 [33]	Turkey	Case–control study	75 (50/25)	-	36.8 ± 8.2	Venous blood, skin biopsy samples	iNOS, MPO	↑MPO (*p* < 0.05)
21.	Yazici et al. 2016 [34]	Turkey	Case–control study	43 (29/14)	11.4 ± 9.3	39.1 ± 12.4	Venous blood	MPO, PCC, AOPP, LOOH, PP	↑MPO, PCC, AOPP, LOOH, PP (*p* < 0.05)
22.	Zhou et al. 2015 [35]	China	Case–control study	379 (214/165)	-	41.0 ± 12.6	Venous blood	TB, CRP	↓TB (*p* < 0.001) ↑CRP (*p* < 0.001)
23.	Ikonomidis et al. 2015 [36]	Greece	Cross sectional study	158 (118 (59 Ps, 59 CAD)/40)	-	51 ± 12.2	Venous blood	MDA, IL-6	↑MDA, IL-6 (*p* < 0.05)
24.	Surucu et al. 2015 [37]	Turkey	Case–control study	87 (40/47)	10.1 ± 7.7	37.9 ± 10.8	Venous blood	Prolidase, TOS, TAS, OSI	↓TAS (*p* = 0.01) ↑Prolidase, TOS (*p* = 0.01), OSI (*p* < 0.001)
25.	Chandrashekar et al. 2015 [38]	India	Cross sectional study	86 (43/43)	4.1 ± 4	44.6 ± 12.0	Venous blood	25-OH-vitD, CRP, IMA	↓25-OH-vitD (*p* = 0.004) ↑CRP (*p* = 0.002), IMA (*p* < 0.001)
26.	Nemati et al. 2014 [39]	Iran	Case–control study	200 (100/100)	4.5 ± 2.4	35.7 ± 10	Venous blood	PON-1, SOD, CAT, MDA	↓PON-1, SOD, CAT (*p* < 0.05) ↑MDA
27.	Pujari et al. 2014 [40]	India	Case–control study	180 (90/90)	-	20–60	Venous blood	MDA, vitamin E, CAT	↓Vitamin E, CAT (*p* < 0.001) ↑MDA (*p* < 0.001)
28.	Balta et al. 2014 [41]	Turkey	Case–control study	115 (60/55)	7.5 ± 9.1	36.8 ± 12.8	Venous blood	TB, DB, IB, CRP	↓TB (*p* < 0.08), DB (*p* < 0.001) ↑IB (*p* < 0.05), CRP (*p* < 0.001)
29.	Meki et Shobaili, 2014 [42]	Saudi Arabia	Case–control study	80 (60/22)	10.3 ± 0.9	30.2 ± 1.4	Venous blood	NO	↑NO (*p* < 0.001)
30.	He et al. 2014 [43]	China	Cross sectional study	50 (25/25)	-	43.04 ± 11.15	Venous blood	MDA, PON-1	↓PON-1 (*p* < 0.01) ↑MDA (*p* < 0.05)
31.	Kaur et al. 2013 [44]	Estonia	Case–control study	107 (60/47)	18.6 ± 11.0	43.2 ± 12.4	Venous blood	TPX, TAS, OSI, Methylglycoxal	↓TAS (*p* < 0.001) ↑TPX, OSI (*p* < 0.001), Methylglycoxal (*p* = 0.01)
32.	Damasiewicz-Bodzek et Wielkoszynski, 2012 [45]	Poland	Case–control study	160 (80/80)	10.2 ± 8.1	37.1 ± 10.8	Venous blood	AGE, Ab anti CEL, Ab anti CML	↑AGE, Ab anti CEL, Ab anti CML (*p* < 0.05)
33.	Emre et al. 2012 [46]	Turkey	Case–control study	116 (54 (28 smokers, 26 non-smokers)/62)	9.51 ± 7,19 (non-smokers), 9.715 ± 7.84 (smokers)	39.9 ± 11.1 (non-smokers), 39.6 ± 12.9 (smokers)	Venous blood	TOS, TAS, AS, OSI	↓TAS (*p* < 0.01), PON-1 (*p* = 0.01—in smokers) ↑TOS, OSI (*p* < 0.01)
34.	Gabr and Al-Ghadir, 2012 [47]	Egypt	Case–control study	75 (55/20)	4.72 ± 1.7	29 ± 13.6	Venous blood	MDA, NO, SOD, CAT, TAS	↓TAS, SOD, CAT (*p* < 0.001) ↑MDA, NO (*p* < 0.001)
35.	Ozdemir et al. 2012 [48]	Turkey	Case–control study	52 (26/26)	6.6 ± 5.8	38.9 ± 11.5	Venous blood	IMA	↑IMA (*p* = 0.001)
36.	Lima et Kimball, 2011 [49]	USA	Cross sectional study	116 (42/72)	-	49.4	None (non-invasive measurements)	Skin Carotenoid level	↓Skin carotenoid level (*p* = 0.003)
37.	Usta et al. 2011 [50]	Turkey	Cross sectional study	77 (52 (27 without MS, 25 with MS)/25)	10 (without MS), 10 (with MS)	-	Venous blood	TAS, TOS, PON-1, AS	↓PON-1, AS
38.	Ferretti et al. 2011 [51]	Italy	Case–control study	48 (23/25)	-	47.5 ± 13.5	Venous blood	PON-1, AS, LOOH	↓PON-1, AS ↑LOOH (*p* < 0.001)
39.	Sikar Akturk et al. 2011 [52]	Turkey	Case–control study	46 (23/23)	-	42.8 ± 16.5	Venous blood, skin biopsy samples	NO, MDA	↑NO, MDA (*p* < 0.001)
40.	Basavaraj et al. 2011 [53]	India	Case–control study	40 (30/10)	-	25–45	Venous blood	8-OHdG, TAS	↑8-OHdG (*p* < 0.05)
41.	Kadam et al. 2010 [54]	India	Case–control study	120 (90/30)	-	-	Venous blood	MDA, NO, SOD, CAT, TAS	↓TAS, SOD, CAT (*p* < 0.01) ↑MDA, NO (*p* < 0.01)
42.	Abeyakirthi et al. 2010 [55]	Scotland	Case–control study	16 (8/8)	-	21–50	Skin tape strips	Ornithine, Arginine	↑Ornithine (*p* < 0.001) ↓Arginine (*p* = 0.005)
43.	Hashemi et al. 2009 [56]	Iran	Case–control study	86 (40/46)	-	30.6 (8–80)	Venous blood	ADA, s-TIC, TAS	↓TAS (*p* = 0.02) ↑ADA (*p* < 0.001), s-TIC (*p* < 0.001)
44.	Nakai et al. 2009 [57]	Japan	Case–control study	70 (49 (29 Ps, 21 AD)/20)	-	55 (21–76)	Urine	Nitrate, 8-OHdG, MDA	↑Nitrate (*p* = 0.03), 8-OHdG (*p* = 0.03) (Ps versus control)
45.	Toker et al. 2009 [58]	Turkey	Case–control study	53 (30/23)	7.2	30.4 ± 10.6	Venous blood	MDA, TAS, PON-1, AS	↑PON-1, sodium stimulated PON-1 (*p* < 0.05), AS (*p* < 0.01)
46.	Kaur et al. 2008 [59]	Estonia	Case–control study	44 (22/22)	-	48	Venous blood	Adiponectin, GSH, GSSG	↑GSSG/GSH
47.	Tekin et al. 2007 [60]	Turkey	Case–control study	124 (84/40)	-	39 (17–58)	Skin biopsy samples	Ox-LDL	↑Ox-LDL
48.	Karaman et al. 2007 [61]	Turkey	Case–control study	66 (36/30)	7.08 ± 4.52	39.3 ± 13.4	Venous blood	SOD, GSH-Px(e), CAT	↓SOD (*p* < 0.001), CAT (*p* < 0.05) ↑GSH-Px(e) (*p* < 0.05)
49.	Rocha-Pereira et al. 2004 [62]	Portugal	Case–control study	100 (60/40)	0.5–50	46 ± 12	Venous blood	TBA, TAS, Transferrin, Ceruloplasmin, CRP	↓TAS (*p* < 0.001)
50.	Kural et al. 2003 [63]	Turkey	Case–control study	70 (35/35)	-	27–43	Venous blood	AuAb-oxLDL, CRP, MDA, LOOH, TAS, SOD, GSH-Px, GSH-R, CAT	↓CAT(e)(*p* = 0.001), SOD (*e-p* = 0.01, *p-p* = 0.003), GSH-Px(e) (*p* = 0.02), TAS(p) (*p* = 0.03) ↑AuAb-oxLDL (*p* = 0.002), MDA (from LDL *p* = 0.018, from oxLDL-p < 0.001, from e-p < 0.001), LOOH (*p* = 0.001)
51.	Baz et al. 2003 [64]	Turkey	Case–control study	59 (35/24)	7.83 ± 8.14	42.5 ± 13.7	Venous blood	MDA, SOD, TAS	↓TAS (*p* = 0.001) ↑SOD (*p* = 0.01), MDA (*p* = 0.005)
52.	Yldirim et al. 2003 [65]	Turkey	Case–control study	44 (22/22)	10	37	Venous blood, skin biopsy samples	SOD(e), GSH-Px(e), CAT(p), MDA(p and skin)	↓SOD (e) (*p* < 0.05) ↑CAT(p) (*p* < 0.05), MDA skin (*p* < 0.01)
53.	Relhan et al. 2002 [66]	India	Case–control study	80 (40/40)	5.6	-	Venous blood	MDA, Thiols	↓Thiols (*p* < 0.001) ↑MDA (*p* < 0.001)

CRP—C-reactive protein, CAT—catalase, MPO—myeloperoxidase, FOX—ferroxidase, IMA—ischemia modified albumin, PON-1—paraoxonase 1, TOS—total oxidant status, TAS—total antioxidant status, MDA—malondialdehyde, 8H2D—8-hydroxy 2′-deoxyguanosine, AOPP—advanced oxidation protein products, NADPH oxidase—nicotinamide adenine dinucleotide phosphate, ROS—reactive oxygen species, GSH-Px—glutathione peroxidase, GSH—glutathione, TrxR—thioredoxin reductase, FRAP—ferric reducing ability of plasma, AS—arylesterase, OSI—oxidative stress index, Px—salivary peroxidase, SOD—superoxide dismutase, AGE—advanced glycation end-products, LOOH—lipid hydroperoxides, OxLDL—oxidized low-density lipoproteins, NO—nitric oxide, 8-OHdG—8-hydroxy guanosine, iNOS—inducible nitric oxide, GSH-R—glutathione reductase, MS—metabolic syndrome, 25-OH-vitD—25-hydroxy-vitamin D, ADA—adenosine deaminase, s-TIC—serum trypsin inhibitory capacity, TBA—thiobarbituric acid, AD—atopic dermatitis, TPX—total peroxide concentration, PCC—protein carbonyl compounds, PP—pyrrolized protein, GSSG—oxidized glutathione, HO—hemoxygenase, TB—total bilirubin, DB—direct bilirubin, IB—indirect bilirubin, Cu—copper, Fe—iron, Trf—transferrin, Cp—ceruloplasmin, AuAb-oxLDL—autoantibodies anti-oxidized LDL, SH—thiol, SS—disulfide, CAD—coronary artery disease, IL-6—interleukin-6, Ab anti-CEL—anti-carboxyethyllysine antibodies, Ab anti CML—anti-carbocymethyllysine antibodies. e—erythrocyte, *p*—plasma, s—skin. ↑, increased. ↓, decreased.

**Table 2 antioxidants-11-00282-t002:** Polymorphisms of Genes Encoding Markers or Enzymes of Oxidative Stress in Psoriasis.

Author/Year	Country	Sample (Psoriasis/Controls)	Mean Age of the Group (Years)	Disease Duration (Years)	Samples	Biomarkers Assessed	Genetic Polymorphisms	Results
Hernandez-Collazo et al. 2020 [67]	Mexico	228 (104/124)	48.1 ± 16.0	10 ± 1.7	Venous blood	TC, TG, LDL, HDL, VLDL, AI	PON-1 rs662 (A>G) and rs854560 (A>G)	↓PON-1 and AS activity ↑G allele of rs662 (A > G): risk for psoriasis, T allele of rs854560 (A > T): susceptibility to psoriasis ↑AG haplotype: more frequent in psoriasis (*p* < 0.05) The AA and AG genotypes of rs662 (A > G) and TT and AA genotypes of rs854560 (A > T): ↓PON-1 and AS activity
Guarneri et al. 2019 [68]	Italy	296 (148/148)	53.7 ± 14.9	-	Bucal swabs samples	GST	GSTM1/GSTT1	GSTT1 null (OR = 3.73) and GSTM1/GSTT1 “double null” (OR = 5.94)
Solak et al. 2016 [69]	Turkey	207 (105/102)	44.5 ± 13.2	-	Venous blood	GSTT1, GSTM1	GST	↑GSTT1 similar in psoriasis and controls, but more frequent in the former (*p* = 0.06)
Chang et al. 2015 [70]	Taiwan	792 (280/512)	48 ± 17	-	Buccal swabs, venous blood	iNOS	(CCTTT) n pentanucleotide polymorphisms in promoter region of iNOS gene	Psoriasis less likely in LL genotype carriers versus non-carriers (*p* = 0.03)
Asefi et al. 2014 [71]	Iran	200 (100/100)	35.3 ± 10.9	10.2 ± 5.8	Blood	MDA, lipids, apolipoproteins VAP-1	MTHFR 677-T	Dominant/recessive model (CC + CT/TT) and T allele of MTHFR-677 alleles increased risk of psoriasis (7.45 and 1.76 times) ↑MTHFR-677-T (C/T + T/T) allele: serum MDA, VAP-1, APOB, APOB/APOA1 MTHFR-677-T allele frequencies in psoriasis patients were significantly higher
Asefi et al. 2012 [72]	Iran	200 (100/100)	35.3 ± 10.9	-	Venous blood	AS, MDA, APOB ⁄APOA1, APOB, LP(a)	PON-1 55 Met	PON-1 55 M allele—associated with psoriasis (OR = 1.96, *p* = 0.01) Psoriasis + PON-1 M (M⁄L + M⁄M) allele—higher MDA levels, apolipoprotein B, APOB, lower AS activity
Schnorr et al. 2005 [73]	Germany	20 (10/10)	47 ± 12	-	Skin biopsy samples	-	CATs (CAT-1, CAT-2A, CAT-2B)	CAT-1: upregulated in psoriasis (*p* < 0.05) CAT-2A, CAT-2B: unaltered in psoriatic skin
Vašků et al. 2002 [74]	Czech Republic	272 (130/142)	44 ± 15 years	-	Venous blood	RAGE	G82S, 1704G/T, 2184A/G, 2245G/A	2184A/G allele: more frequent in psoriasis (*p* = 0.001)—G82S, 1704G/T and 2245A/G9 polymorphisms not associated with psoriasis

*Legend:* GST/GSH—glutathione S-transferases; PON-1—paraoxonase 1; MDA—malondialdehyde; AS—arylesterase; APOB—apolipoprotein B; APOA1—apolipoprotein A1; MTHFR—methylentetrahydrofolatereductase, VAP-1—vascular adhesion protein-1; CAT—catalase; SOD1- superoxide dismutase 1; SOD2—superoxide dismutase 2; NFE2L2—nuclear factor, erythroid 2 like 2; CYBB—cytochrome b-245 beta chain; SO—superoxide; IL-17—interleukin-17; TAC—total antioxidant capacity; LP(a)—lipoprotein-a; TC—total cholesterol; LDL—low-density lipoprotein; HDL—high-density lipoprotein; CVD—cardiovascular disease TG—Triglycerides; VLDL—very low-density lipoprotein; AI—atherogenic index (TC/HDL-C ratio); iNOS—inducible nitric oxide synthase; TNF-α—tumor necrosis factor-α; IL-10—interleukin-10; N-AT—N-acetyltransferase; eNOS—endothelial nitric oxide synthase; CATs—cationic amino acid transporters, RAGE—receptor for advanced glycation end-products, GST—glutathione S-transferase. ↑, increased. ↓, decreased.

**Table 3 antioxidants-11-00282-t003:** The effect of anti-psoriasis therapy on oxidative stress markers.

	Author/Year	Country	Intervention	Treatment Duration	Measured Parameters	Samples	Study Group (Psoriasis/Controls)	Mean Age of Patients	Effect on the Measured Parameters
1	Akbulak et al. 2017 [75]	Turkey	MTX 10–15 mg/week	≥12 weeks	expression of GST and CYP enzymes	Skin biopsy samples	43 (21/22)	42.5 ± 10.9	↑GSTK1, GSTM1, GSTT1, CYP1B1, CYP2E1: in the psoriasis tissues (*p* < 0.05); No significant decrease after MTX treatment
2	Elango et al. 2013 [76]	India	MTX 7.5 mg/week	12 weeks	ROS, MDA, nitrate, SOD, CAT, TAS	Venous blood, skin biopsy samples	103 (58/45)	46.4 ± 14.1	↑ROS (lesional skin), MDA (serum) After 6 and 12 weeks of treatment (*p* < 0.001) ↓Serum nitrite, SOD, TAS, CAT (*p* < 0.001) (versus controls) (no difference after therapy)
3	Kılıc et al. 2013 [77]	Turkey	MTX	8 weeks	TAS, TOS, OSI, PON-1	Venous blood	26 (26/0)	45.3 ± 11.7	No significant differences pre and post-treatment
4	Tekin et al. 2006 [78]	Turkey	MTX 20 mg/week	Until the disappearance of the lesions	Nitrite-nitrate	Venous blood	43 (22/21)	35.0 ± 11.8	↓Nitrite-nitrate (*p* < 0.05)
5	Darlenski et al. 2021 [79]	Bulgaria	NB-UVB 311 nm	10 sessions	PASI, DLQI, skin carotenoid levels	Non-invasive	29 (20/9)	48.9	↓PASI, DLQI (*p* < 0.001), carotenoid levels (*p* > 0.05)
6	Darlenski et al. 2018 [80]	Bulgaria	NB-UVB 311 nm	14 sessions	MDA, ROS, Asc, CAT	Venous blood	47 (22/25)	50.9	↓ROS, Asc, MDA, CAT (*p* < 0.001)
7	Wacewicz et al. 2017 [81]	Poland	NB-UVB	20 sessions	Se, Zn, Cu, Cu/Zn, CRP, TAS	Venous blood	118 (60/58)	41.2 ± 12.5	↓Se and TAS (*p* < 0.05) Cu/Zn ratio, Cu, Zn—no changes after NB-UVB
8	Karadag et al. 2016 [82]	Turkey	NB-UVB 311 nm	20–36 sessions	GST, CYP	Skin biopsy samples	54 (32/22)	37.2 ± 14.8	GST1K1, GST1M1, GST1O1, GST1T1, CYP1A1, CYP1B1, CYP2E1: no differences pre and post-treatment
9	Pektas et al. 2013 [83]	Turkey	NB-UVB 310–315 nm	30 sessions	hsCRP, TAS, TOS, OSI, PON-1, ARE	Venous blood	24 (24/0)	37.9 ± 12.3	↓PASI (*p* = 0.001) ↑TOS, OSI (*p* < 0.001)
10	Coimbra et al. 2012 [84]	Portugal	NB-UVB: 17 pts; PUVA: 20 pts; calcipotriol: 10 pts	12 weeks	TB, MBH, MPB3, TAS, TBA, elastase, lactoferrin, CRP	Venous blood	113 (73/40)	45 ± 15	PUVA: ↓leukocytes, neutrophils, elastase, lactoferrin, CRP, TBA, TBA/TAS NB-UVB:↓elastase, lactoferrin, CRP, TBA, ↑TBA/TAS; MPB3 monomers PUVA + NB-UVB: changed MPB3 profile
11	Karaarslan et al. 2006 [85]	Turkey	BB-UVB	21 weeks	TBARS, nitrite-nitrate	Venous blood	52 (32/20)	42.0 ± 11.1	↑TBARS (*p* < 0.05), nitrite-nitrate levels (*p* < 0.01) negative correlation total nitrite—TBARS levels post-treatment (r = −0.58, *p* = 0.03)
12	Barygina et al. 2013 [86]	Italy	infliximab 5 mg/kg every 8 weeks	6 months	ROS, GSH, NADPH oxidase, PCO, MDA, TAS, TBARS (lipid peroxidation)	Venous blood	47 (29/18)	47 ± 8	↓PCO, TBARS, TBARS, ROS (*p* < 0.05) ↑TAS (*p* < 0.05)
13	Wolk et al. 2017 [87]	USA	tofacitinib 5 mg/10 mg twice daily	16 weeks	HDL, LDL, PON-1, LCAT, SA-A, hsCRP	Venous blood	161 (70 tofacitinib 5 mg, 71 tofacitinib 10 mg, 50 placebo)	42.3–50.9	↑LDL, HDL (*p* < 0.05); TC/HDL: remained constant; PON-1, LCAT (*p* < 0.05) compared with placebo; ↓SA-A, hsCRP (*p* < 0.05) compared with placebo.
14	Pastore et al. 2011 [88]	Italy	efalizumab 1 mg/kg/week	12 weeks	Nitrites-nitrates, MDA, TBARS, SOD, Cu, Zn, GST, CAT, acrolein-protein adducts, 9.4-HNE, SF, GF, PUFA.	Venous blood	50 (26/24)	42.9	pro-inflammatory cytokines, PUFAs esterified in phospholipids of RBC membranes were not affected. ↓Nitrites–nitrates, MDA levels, CAT (in non-responders) ↑GPx, GST (in non-responders)
15	Campanati et al. 2012 [89]	Italy	etanercept or adalimumab	12 weeks	iNOS, TNF-alpha, VEGF, NO, SOD, CAT, GST, GSH	Skin biopsy samples	12 (6/6)	51 ± 5.8 and 52 ± 6.9	↓VEGF (*p* < 0.05, regardless of the treatment) NO (*p* < 0.05, regardless of the treatment) CAT (*p* < 0.05, for adalimumab in non lesional and perilesional skin and for etanercept in non lesional skin) GST (*p* < 0.05, regardless of the treatment for perilesional skin) ↑SOD (*p* < 0.05, for adalimumab in perilesional and lesional skin) GSH (*p* < 0.05, for adalimumab in non lesional skin)

*Legend:* MTX—Methotrexate; ROS—reactive oxygen species; MDA—malondialdehyde; SOD—superoxide dismutase; CAT—catalase; TAS—total antioxidant status; Asc—ascorbyl radicals; PASI—psoriasis area and severity index, DLQI—dermatology life quality index; NB-UVB—narrow band ultraviolet B; TAC—total antioxidant capacity; WBC—white blood cells, PCO—protein carbonyl content; GSH—glutathione content; TBARS—thiobarbituric acid reactive substances; hsCRP—high sensitive C-reactive protein; TOS—total oxidant status; OSI—oxidative stress index; PON-1—serum paraoxonase-1; AS—arylesterase; TC—total cholesterol; LDL—low-density lipoprotein; HDL—high-density lipoprotein; ALT—alanine aminotransferase; TSH—thyroid-stimulating hormone; FAE—fumaric acid esters; GSTs—Glutathione S-transferases; AE—adverse effects, CRP—c-reactive protein, Cu—copper, Se—selenium, Zn—zinc, CBD—cannabidiol, GT—Goeckerman therapy (combined exposure of 3% crude coal tar ointment and UV radiation), BPDE—benzo[a]pyrene-7,8-diol-9,10-epoxide, BB-UVB—broad-band ultraviolet B, TBARS—thiobarbituric acid reactive substance, 4-HNE—4-hydroxy nonenal-protein, SF—stimulating factors, GF—growth factors, PUFA—polyunsaturated fatty acids, GPx—glutathione, peroxidase, l-NMMA—NG-monomethyl-l-arginine, PUVA—psoralen plus UVA, TBA—thiobarbituric acid, CRP—C-reactive protein, MBH—membrane-bound hemoglobin, TB—total bilirubin, MPB3—membrane protein band 3, MCV—mean cell volume, MCH—mean cell hemoglobin, LCAT—lecithin-cholesterol acyltransferase, SA-A—serum amyloid A, CYP—cytochrome p450, 8H2DG—8-hydroxy-2′-deoxyguanosine, 8HG—8-hydroxyguanosine, 8HGN—8-hydroxyguanine. ↑, increased. ↓, decreased.

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
