# Peer review of "The Involvement of Oxidative Stress in Psoriasis: A Systematic Review"

_antioxidants, 2022, doi:10.3390/antiox11020282_

Round 1

Reviewer 1 Report

In this systematic review Dobrica et al. examine the literature suggesting that oxidative stress might play a role in psoriasis. They focus on studies performed in human patients and examine also potential effects of therapies on parameters related to oxidative stress and antioxidant protection. The review is timely and addresses an important topic. Suggestions for improvement are provided below.

(1) The authors must be cautious in stating that many of the studies discussed indicate that oxidative stress plays a role. Most of the studies described only show associations between particular parameters and psoriasis. For example, on page 10 (lines 196-200) the authors state that the observation that there are lower levels of total bilirubin and direct bilirubin in psoriasis emphasizes “once again the role of TB in the systemic complications of psoriasis…” but really these represent only associations. Without some way of manipulating these levels and showing an effect on the indicated parameter, it is not possible to say definitively that they play a role in the disease or its complications. Likewise on page 11, line 234, there is only an association between carbonyl levels and psoriasis.

(2) The authors use an excessive number of abbreviations. While this seems reasonable for the Tables with their limited space and for which the definitions can be included as footnotes, it makes it difficult for the reader to comprehend the information in the
text. It is suggested that in the text the majority of the abbreviations be spelled out.

(3) Pruritus only affects about half of patients with psoriasis; therefore, the description in line 20 of the abstract and line 44 of the Introduction seems inaccurate.

(4) In lines 21 of the abstract and line 45 of the Introduction rather than “extension” it should be “extensor.”

(5) The up and down arrows in the tables are often not appropriately located (often overlapping words). Also, “xanthine” has an “h” and “Scotland” in Table 1 does not have an “h.”

(6) It is “advanced glycation end-products” or “AGE” and not “advanced end glycation products” nor “AEG.”

(7) On page 10, lines 228-230, how were AGE measurements made “in the absence of integument”? This sentence is unclear.

(8) On page 13, line 368, what does “with STEPS over” mean? This sentence is unclear.

(9) Some of the changes seen with the treatment of psoriasis seem paradoxical and are not well discussed. Indeed, the authors could do a much better job of synthesizing the information presented in general.

(10) NFkappaB tends to be a pro-inflammatory transcription factor and inflammation usually increases reactive oxygen species generation and oxidative stress. Therefore, it is unclear why activation of NFkappaB would promote antioxidant systems. What is the evidence for this in Figure 2?

(11) On page 24, line 631-632, what does it mean if the indicated antioxidant enzyme systems are “more frequently encountered in psoriasis”?

Author Response

Dear Academic Editor,

Dear Peer-Reviewers,

We are very thankful to you for the pertinent notes; we have carefully read the comments and have revised/completed the manuscript accordingly. Our responses are given in a point-by-point manner below, as well, all the changes to the manuscript are highlighted in yellow.

We hope that in this new form, the manuscript will be suitable for publication in Antioxidants.

Reviewer 1

We would like to thank you for your valuable comments which helped us improve the manuscript. All suggestions were taken into consideration and appropriate information, as well as required corrections, were provided. New/corrected parts are highlighted in yellow to facilitate the assessment of changes. We did our best to fulfill the expectations and we hope that you will be satisfied with our corrections.

In this systematic review Dobrica et al. examine the literature suggesting that oxidative stress might play a role in psoriasis. They focus on studies performed in human patients and examine also potential effects of therapies on parameters related to oxidative stress and antioxidant protection. The review is timely and addresses an important topic. Suggestions for improvement are provided below.

Response: Thank you for your positive feedback regarding our manuscript.

 (1) The authors must be cautious in stating that many of the studies discussed indicate that oxidative stress plays a role. Most of the studies described only show associations between particular parameters and psoriasis. For example, on page 10 (lines 196-200) the authors state that the observation that there are lower levels of total bilirubin and direct bilirubin in psoriasis emphasizes “once again the role of TB in the systemic complications of psoriasis…” but really these represent only associations. Without some way of manipulating these levels and showing an effect on the indicated parameter, it is not possible to say definitively that they play a role in the disease or its complications. Likewise on page 11, line 234, there is only an association between carbonyl levels and psoriasis.

Response: Thank you for pointing this out. Indeed, the discussed papers mostly highlighted the presence of associations between particular oxidative stress parameters and psoriasis. Thus, we have revised the text In order to emphasize this aspect and avoid overstatements.

(2) The authors use an excessive number of abbreviations. While this seems reasonable for the Tables with their limited space and for which the definitions can be included as footnotes, it makes it difficult for the reader to comprehend the information in the text. It is suggested that in the text the majority of the abbreviations be spelled out.

Response: Thank you for pointing this out. We have decided to avoid the use of abbreviations and have kept only the most commonly employed ones.

(3) Pruritus only affects about half of patients with psoriasis; therefore, the description in line 20 of the abstract and line 44 of the Introduction seems inaccurate.

Response: Thank you for pointing this out. Indeed, psoriasis is not pruritogenic in the majority of cases. We have revised the aforementioned lines of the paper to avoid this inaccurate statement.   

(4) In lines 21 of the abstract and line 45 of the Introduction rather than “extension” it should be “extensor.”

Response: Thank you for pointing this out. We have corrected this mistake.

(5) The up and down arrows in the tables are often not appropriately located (often overlapping words). Also, “xanthine” has an “h” and “Scotland” in Table 1 does not have an “h.”

Response: Thank you for pointing this out. We have corrected the issue related to the placement of arrows. In addition, we have corrected the spelling mistakes.

(6) It is “advanced glycation end-products” or “AGE” and not “advanced end glycation products” nor “AEG.”

Response: Thank you for pointing this out. We have corrected this mistake.

(7) On page 10, lines 228-230, how were AGE measurements made “in the absence of integument”? This sentence is unclear.

Response: Thank you for pointing this out. We replace the "presence or absence of the integument" with "the presence or absence of lesions on the integument". We employed "integument" as a synonym of "skin" to avoid the repetitive use of this word.

(8) On page 13, line 368, what does “with STEPS over” mean? This sentence is unclear.

Response: Thank you for pointing this out. This was indeed an error. We have replaced “with STEPS over” with “with PASI over 6 compared with those with PASI under 6”.

(9) Some of the changes seen with the treatment of psoriasis seem paradoxical and are not well discussed. Indeed, the authors could do a much better job of synthesizing the information presented in general.

Response: Thank you for pointing this out. We have amended this section of the manuscript and have further discussed these aspects.

(10) NFkappaB tends to be a pro-inflammatory transcription factor and inflammation usually increases reactive oxygen species generation and oxidative stress. Therefore, it is unclear why activation of NFkappaB would promote antioxidant systems. What is the evidence for this in Figure 2?

Response: Thank you for pointing this out. Indeed, the figure was not very clear and we have amended it. The two pathways, i.e., NRF2 and NF-kB, have opposite effects. NRF2 is stimulated by ROS to induce antioxidant enzymes, whereas NF-kB stimulated the generation of ROS. These mechanisms are explained in detail in reference [93]. Nevertheless, we have reorganized the figure in order to convey this message more clearly.

(11) On page 24, line 631-632, what does it mean if the indicated antioxidant enzyme systems are “more frequently encountered in psoriasis”?

Response: Thank you for pointing this out. We wanted to emphasize the fact that some genes are more frequently expressed in psoriasis versus the general population. We have reworded the statement from "more frequently encountered" to "more frequently expressed".

Thank you for taking the time to review our paper and for your valuable suggestions. Overall, we believe that thanks to your pertinent comments, the quality of the manuscript has been improved.

Cordially yours,

The authors.

Reviewer 2 Report

The study is a very interesting review of oxidative stress markers in psoriasis. Tables are very informative and are very helpful in the results’ presentation. Most of data are repeated in the tables and in the text, what is in my opinion unnecessary. If you decided to present the results both in the tables and in the text, it would be better to describe together the results of the same marker measured by the different authors. Some results of the measured parameters are not involved in the tables (i.e. Table 1, point 3 (Px, CAT, SOD), point 5 (CRP), point 33 (PON1). I think all results should be included, especially if they are statistically significant. There is a mistake in the name of the author in the 2. point of table1 (Oszukawska). The different abbreviations are used for the same marker – PON-1, POX-1, AS for paraoxonase/arylesterase. In most of the papers PON-1 is used for paraoxonase. You should consider to unify the abbreviations in the paper with the data from the literature. Tables 2 and 3 are not well formatted and arrows are in the wrong positions.

Author Response

Dear Academic Editor,

Dear Peer-Reviewers,

We are very thankful to you for the pertinent notes; we have carefully read the comments and have revised/completed the manuscript accordingly. Our responses are given in a point-by-point manner below, as well, all the changes to the manuscript are highlighted in yellow.

We hope that in this new form, the manuscript will be suitable for publication in Antioxidants.

Reviewer 2

We would like to thank you for your valuable comments which helped us improve the manuscript. All suggestions were taken into consideration and appropriate information, as well as the required corrections, were provided. New/corrected parts are highlighted in yellow to facilitate the assessment of changes. We did our best to fulfill the expectations and we hope that you will be satisfied with our corrections.

The study is a very interesting review of oxidative stress markers in psoriasis. Tables are very informative and are very helpful in the results’ presentation.

Response: Thank you for your positive feedback regarding our manuscript.

Most of data are repeated in the tables and in the text, what is in my opinion unnecessary. If you decided to present the results both in the tables and in the text, it would be better to describe together the results of the same marker measured by the different authors.

Response: Thank you for pointing this out. This would have been a better option indeed. However, most of the analyzed markers were evaluated in one study only and thus grouping them was nearly impossible. We will discuss the opportunity to include some of the potentially redundant information in a Supplementary file to improve the readability of the paper.

Some results of the measured parameters are not involved in the tables (i.e. Table 1, point 3 (Px, CAT, SOD), point 5 (CRP), point 33 (PON1). I think all results should be included, especially if they are statistically significant.

Response: Thank you for pointing this out. We have completed the table with the missing data that was statistically significant. We did not include the results which did not reach statistical significance in order to increase the readability of the tables, however, these findings are reported in the text.

There is a mistake in the name of the author in the 2. point of table1 (Oszukawska).

Response: Thank you for pointing this out. We have corrected the name of the author.

The different abbreviations are used for the same marker – PON-1, POX-1, AS for paraoxonase/arylesterase. In most of the papers PON-1 is used for paraoxonase. You should consider to unify the abbreviations in the paper with the data from the literature.

Response: Thank you for pointing this out. We have performed the suggested corrections and unified the abbreviations.

Tables 2 and 3 are not well formatted and arrows are in the wrong positions.

Response: Thank you for pointing this out. We have amended the placement of arrows and have reformatted the tables.

Thank you for taking the time to review our paper and for your valuable suggestions. Overall, we believe that thanks to your pertinent comments, the quality of the manuscript has been improved.

Cordially yours,

The authors.
